# Phlegmonous Appearance in the Ipsilateral Paracardiac Fat without Paracardiac Lymph Node Enlargement on Chest CT Favors the Diagnosis of Pleural Tuberculosis over Malignant Pleural Effusion

**DOI:** 10.3390/diagnostics10121041

**Published:** 2020-12-03

**Authors:** Dongjun Lee, Min Ji Son, Seung Min Yoo, Hwa Yeon Lee, Charles S. White

**Affiliations:** 1Military Service in Korean Army, Hongcheon 25117, Korea; sirious0416@naver.com; 2Department of Radiology, CHA University Bundang Medical Center, Seongnam 13497, Korea; smj3006@naver.com; 3Smile Radiologic Clinic, Seoul 33066, Korea; hynlee1@hanmail.net; 4Department of Radiology, University of Maryland, Baltimore, MD 21201, USA; cswhite999@gmail.com

**Keywords:** pleural tuberculosis, malignant pleural effusion, paracardiac fat, chest CT

## Abstract

This study investigated the potential role of paracardiac fat stranding (FS) interspersed with multiple fluid collections (FC) as a clue to differentiate between pleural tuberculosis (pleural TB) and malignant pleural effusion (MPE). The authors retrospectively analyzed chest computed tomography (CT) findings of 428 patients, 351 with pleural TB and 77 with MPE, focusing on the paracardiac fat, and level of pleural adenosine deaminase (ADA) and blood C-reactive protein (CRP). Two radiologists independently evaluated the chest CT findings regarding the paracardiac fat pad ipsilateral to the effusion, including FS, FC, phlegmonous appearance (a combination of the FS and multiple FC), and the presence of lymph node enlargement (>1 cm in short axis diameter). There were significant differences between patients with pleural TB and those with MPE with respect to the prevalence of phlegmonous appearance in the ipsilateral paracardiac fat (47.6% and 10.4%, *p* < 0.001, OR = 7.8; 95% CI 3.7–16.8) and paracardiac lymph node enlargement (1.4% and 19.5%, *p* < 0.001, OR = 0.06; 95% CI 0.02–0.2) on CT. In contrast, there was no difference in the prevalence of isolated FS or multiple FC within the ipsilateral paracardiac fat between the two groups. Median pleural ADA and serum CRP level were higher in patients with pleural TB accompanied by phlegmonous appearance in paracardiac fat compared to those without that appearance (ADA: median 104 IU/L versus 90 IU/L, *p* < 0.001; CRP: 6.5 mg/dL versus 4.2 mg/dL, *p* < 0.001). In conclusion, phlegmonous appearance in the ipsilateral paracardiac fat without paracardiac lymph node enlargement on chest CT favors a diagnosis of pleural TB over MPE.

## 1. Introduction

Although the presence of nodular pleural thickening (>1 cm) on computed tomography is reported to be a specific CT finding of malignant pleural effusion (MPE), the finding is often absent [1,2]. In many patients lacking prominent nodular pleural enhancement on chest CT, the differentiation of pleural tuberculosis (TB) from MPE can be a challenging task for interpreting physicians [1,3,4,5,6]. Lymphatic channels within the ipsilateral paracardiac fat can serve as a route for drainage of excess pleural effusion or spread of pleural inflammation or malignancy [7,8]. However, the presence of paracardiac abnormalities ipsilateral to an indeterminate pleural effusion on chest CT such as fat stranding (FS), fluid collection (FC), phlegmonous appearance (a combination of the FS and multiple FC), and paracardiac lymph node enlargement as a means of differentiating between the two entities has not been reported. Thus, the purpose of this study was to evaluate whether the appearance of the ipsilateral paracardiac fat pad on CT has a potential role in distinguishing pleural TB and MPE.

## 2. Materials and Methods

### 2.1. Patient Selection

The institutional review board approved this study and informed consent was waived (IRB-2019-10-078; 21 April 2019). From September 2008 to March 2020, all patients with pleural effusion who underwent contrast enhanced chest CT subsequently confirmed to be due to either pleural TB or MPE were enrolled in this study (n = 800 patients, pleural TB = 652 and MPE = 148). We excluded patients with conditions that could potentially affect interpretation of paracardiac fat on chest CT, including those with previous thoracentesis (n = 187 in pleural TB, n = 35 in MPE), chest tube insertion prior to chest CT scan (n = 34 in pleural TB, n = 14 in MPE), and previous history of thoracic surgery (n = 72 in pleural TB, n = 12 in MPE). In addition, we excluded patients (n = 8 in pleural TB, n = 10 in MPE) who had a pericardial effusion, because effusion-related changes in the pericardium may affect paracardiac fat morphology. There were no patients who received radiation therapy that could affect paracardiac fat attenuation. In addition, antibody test for the human immunodeficiency virus was negative for all enrolled patients. Using these criteria, 301 cases were excluded among the 652 patients with pleural TB, and 71 cases were excluded from among 148 patients with MPE. Ultimately, 351 patients with pleural TB and 77 patients with MPE were included in the study (Figure 1). A diagnosis of pleural TB was established based on a positive acid-fast bacillus stain or culture of the pleural fluid (n = 42), or if there was a lymphocyte-dominant pleural effusion with pleural adenosine deaminase (ADA) level > 40 IU/L (n = 309), followed by resolution of the pleural effusion after anti-tuberculous medication [1,6,7,8]. A diagnosis of MPE was made based on video-assisted thoracoscopic surgery biopsy (n = 40) or pleural biopsy (n = 37).

### 2.2. Image Acquisition

Contrast enhanced chest CT scans were performed using 64-slicemulti-detector CT (MDCT) scanners (Light-speed VCT, GE HealthCare, Milwaukee, WI, USA). Scanning parameters for chest CT were as follows: 120 kV, 200 mA, 0.625 mm collimation, 1.5 mm increment, 3 mm reconstruction. A volume of 60 to 120 mL of intravenous Ioversol (Optiray 320 mg/mL, Tyco Healthcare, Montreal, QC, Canada) was injected based on the patient’s body mass index. CT scanning was started following a 60–80 s delay after contrast administration (bolus-tracking). The scan range for chest CT was from the lower half of the neck to the adrenal glands.

### 2.3. Image Interpretation and Data Evaluation

The CT findings of the 428 consecutive patients were analyzed retrospectively by two radiologists (26 and 7 years of experience in chest CT interpretation). The two radiologists independently assessed the following CT findings on both the axial and coronal CT images in the ipsilateral paracardiac fat: fat stranding, fluid collection, combination of fat stranding and fluid collection, and lymph node enlargement (>1 cm in short axis diameter). We also evaluated the prevalence of the above CT findings in the ipsilateral paracardiac fat in the subgroup of patients with pleural TB and MPE lacking nodular pleural thickening >1 cm (thickening > 1 cm being a classic CT finding favoring MPE). Decisions on each CT finding were reached in a consensus manner if there was disagreement.

Fat attenuation with thin vessels can often be identified in the normal paracardiac area on chest CT (Figure 2a). Abnormal paracardiac fat stranding was defined as the presence of asymmetrical linear densities on the side ipsilateral to the pleural effusion, unaccompanied by tapering and branching ends, which would be more typical of normal paracardiac small vessels (Figure 2b). A paracardiac fluid collection was defined as an area of >1 cm^2^ with fluid attenuation (Hounsfield Unit = 0–30 on precontrast CT) (Figure 2c). An arbitrary low threshold valve of 1 cm^2^ was selected to avoid confusion between a subtle FC and prominent linear densities such as thick FS. Phlegmonous appearance in paracardiac fat was defined as the simultaneous presence of multifocal FS interspersed with multifocal FC in the paracardiac area (Figure 2d,e). We used the term “phlegmonous appearance” in paracardiac fat in this study due to the similarity in the CT findings of periappendiceal phlegmon in patients with appendicitis [9]. An enlarged paracardiac lymph node was considered to be present for lymph node >1 cm in short axis diameter. With respect to tuberculous immune response, we hypothesized that the extent of paracardiac fat stranding and fluid collection in the ipsilateral paracardiac fat might have a positive correlation with the strength of the immune response or inflammation within the pleural effusion. To evaluate such a relationship in pleural TB patients, we investigated whether there is a correlation between the level of pleural ADA value and serum CRP value, and the prevalence of phlegmonous appearance, or FS in the ipsilateral paracardiac fat in patients with pleural TB. Among 351 patients in pleural TB, 322 patients had available laboratory results for pleural fluid ADA level. In addition, 334 patients with pleural TB had an available blood CRP level.

Multiple nodular areas of pleural thickening with contrast enhancement (black and white arrowheads) are noted in the right pleural effusion (asterisks) on a coronal CT image at the level of the right ventricle. Note multiple enlarged lymph nodes measuring >1 cm (white arrows), but absence of phlegmonous appearance in the ipsilateral paracardiac fat.

### 2.4. Statistical Analysis

Statistical analysis was performed using software (SPSS Inc., Chicago, IL, USA). A Chi-square test was performed for categorical variables and student *t*-test or Mann-Whitney U test was performed for continuous variables, respectively. Interobserver agreement for the presence or absence of fat stranding, fluid collection, and phlegmonous appearance in the paracardiac fat ipsilateral to the effusion on CT was evaluated using Cohen’s Kappa coefficient. A statistically significant difference was defined as *p* value < 0.05.

## 3. Results

### 3.1. Characteristics of Patients

The baseline characteristics of the enrolled patients are shown in Table 1.

### 3.2. Prevalence of Phlegmonous Appearance, Fat Stranding, Multiple Fluid Collections, and Lymph Node Enlargement in Paracardiac Fat

The prevalence of phlegmonous appearance, fat stranding, multiple fluid collections, and lymph node enlargement in ipsilateral paracardiac fat in both pleural TB and MPE on chest CT is summarized in Table 2.

Paracardiac fat phlegmonous appearance was observed in 47.6% of patients (167/351) with pleural TB and 10.4% of patients (8/77) with MPE (OR 7.8; 95% CI 3.7–16.8; *p* < 0.001), respectively. Lymph node enlargement in the paracardiac fat was observed in 1.4% patients (5/351) with pleural TB, and in 19.5% patients (15/77) with MPE (OR = 0.06; 95% CI 0.02–0.2; *p* < 0.001) (Figure 3). Notably, there was also a significant difference between pleural TB and MPE in the prevalence of phlegmonous appearance without lymph node enlargement (47.0% (165/351) versus 1.3% (1/77), respectively, *p* < 0.001, OR = 67.4; 95% CI 9.3–490.2). In contrast, there was no difference between the two groups in the prevalence of isolated fat stranding or fluid collection within the ipsilateral paracardiac fat (Table 2). Nodular pleural enhancement >1 cm was found in 1.1% patients (4/351) with pleural TB, and 41.6% patients (32/77) with MPE (*p* < 0.001). In a subgroup analysis excluding patients with nodular pleural enhancement >1 cm, there were significant differences between pleural TB and MPE in the prevalence of a phlegmonous appearance and lymph node enlargement in the ipsilateral paracardiac fat (47.8% (166/347) versus 8.9% (4/45), *p* < 0.001; 1.4% (5/347) versus 8.9% (4/45), *p* = 0.002, respectively). In particular, a phlegmonous appearance in the ipsilateral paracardiac fat without lymph node enlargement was found in 47.3% of patients (164/347) with pleural TB, and 2.2%of patients (1/45) with MPE, respectively (OR 39.4; 95% CI 5.4–289.4; *p* < 0.001) (Table 3).

Interobserver agreement between the two readers regarding the presence or absence of fat stranding, fluid collection, and phlegmonous appearance in the ipsilateral paracardiac fat was excellent (kappa value; 0.91, 0.91, and 0.86, respectively, *p* < 0.001).

### 3.3. Differences of Pleural ADA and Serum CRP Level According to Presence of Phlegmonous Appearance in Pleural TB Patients

There were significant differences in the median value of pleural ADA (104 IU/L versus 90 IU/L, *p* < 0.001) and serum CRP (6.5 mg/dL versus 4.2 mg/dL, *p* < 0.001) between the patients with ipsilateral paracardiac phlegmonous appearance and those who lacked this finding in pleural TB. There was also a significant difference in the level of serum CRP between patients with pleural TB who had ipsilateral paracardiac phlegmonous appearance and those with isolated fat stranding (6.5 mg/dL versus 4.3 mg/dL, *p* = 0.022). In contrast, there was no significant difference in the median value of pleural ADA and serum CRP in other groups (Table 4 and Table 5).

## 4. Discussion

Pleural TB remains a major public health problem in many countries [1,10,11,12]. Various studies have made attempts to differentiate pleural TB and MPE based on clinical and laboratory results [3,10,13,14,15,16]. In contrast, there are fewer studies that attempt to make such a distinction based on CT findings [1,2,17,18].

Previous studies [1,3,17] have suggested that pleural/fissural nodules, circumferential pleural thickening >1 cm pleural thickening, and mediastinal pleural involvement are more frequently found in patients with MPE compared to those with pleural TB. Kim et al. reported that 72.5% of malignant pleural effusions showed one or more of these findings, but 43.7% with pleural TB also showed one or more of the findings [1]. The most specific CT finding to differentiate pleural TB from MPE was nodular pleural thickening >1 cm [1]. However, this finding demonstrated low sensitivity. Thus, differentiation of pleural TB from MPE on CT may not be possible in a substantial proportion of cases due to the low sensitivity of the classically described findings.

Porcel et al. proposed a scoring system for discriminating MPE from benign causes of pleural effusion (including both transudate and exudative pleural effusions) [3]. They established a scoring system using seven factors: any pleural lesion ≥1 cm, liver metastases, an abdominal mass, lung mass or lung nodule ≥1 cm, absence of pleural loculation, absence of pericardial effusion, and a nonenlarged cardiac silhouette. However, fat stranding with multiple FC or lymph node enlargement in the ipsilateral paracardiac fat was not considered in that study [3]. In addition, their results may not be helpful in the sizeable number of patients with undifferentiated pleural effusion who lack nodular pleural enhancement or a primary malignant focus.

A key finding of our study is that CT findings of phlegmonous appearance in combination with a lack of lymph node enlargement in the ipsilateral paracardiac fat appear to be specific for pleural TB. Conversely, we found that the absence of a phlegmonous appearance and the presence of lymph node enlargement over 1 cm in the ipsilateral paracardiac fat strongly suggest the possibility of MPE.

In a subgroup analysis of pleural TB patients lacking nodular pleural enhancement >1 cm on CT, the presence of phlegmonous appearance in the ipsilateral paracardiac fat had a high odds ratio compared to absence of this finding (9.4). In addition, the absence of lymph node enlargement in the ipsilateral paracardiac fat favored a diagnosis of pleural TB in this subgroup. Thus, this combination may be helpful to differentiate pleural TB from MPE in patients lacking nodular pleural enhancement.

Pleural TB is thought to be due to the rupture of a subpleural focus of pulmonary disease into the pleural space with delayed hypersensitivity reaction to mycobacteria or mycobacterial antigens in sensitized individuals [19]. Based on this hypothesis, we supposed that differences in the intensity of immune or inflammatory reaction might influence the presence or absence of fat stranding or fluid collection in the ipsilateral paracardiac fat attenuation.

This study showed that the prevalence of the paracardiac fat stranding with multiple fluid collections, namely phlegmonous appearance, is more frequent in patients with pleural TB than in those with MPE. This result suggests that phlegmonous appearance may represent an extension of the inflammatory process into the paracardiac fat ipsilateral to the effusion. However, some patients with MPE also showed phlegmonous appearance within the ipsilateral paracardiac fat. A possible explanation for this finding is an exaggerated pleural inflammation secondary to a coexistent pulmonary inflammation such as pneumonia. In addition, the possibility of lymphangitic spread or direct tumor invasion cannot be excluded as a cause. However, the presence of phlegmonous appearance unaccompanied by a lymph node enlargement in the ipsilateral paracardiac fat on chest CT favored the diagnosis of pleural TB with high odds ratio (67.4 [9.3–490.2], *p* < 0.001).

We assumed that a stronger immune reaction would lead to an increase in the extent of ipsilateral paracardiac fat stranding with multiple fluid collections. We found a positive correlation between the presence of ipsilateral paracardiac phlegmonous appearance, and the level of pleural ADA and serum CRP in patients with pleural TB. In assessing an analogous situation, a previous study investigated potential factors associated with periappendiceal fat stranding on CT in acute appendicitis [20]. That study suggested that an elevated CRP level is a significant factor associated with the presence of periappendiceal fat stranding on CT [20]. Similarly, we hypothesize that the stronger immune reaction in patients with pleural TB may trigger a prominent inflammatory change within the ipsilateral paracardiac fat, leading to the phlegmonous sign of this study.

However, it is unclear why the prevalence of enlarged lymph nodes over 1 cm in the ipsilateral paracardiac fat is higher in patients with MPE compared to those with pleural TB in this study. Further studies are required to address this issue. This study has limitations. First, no histopathologic confirmation of paracardiac fat attenuation change was obtained. Second, the identification of phlegmonous appearance in the paracardiac fat on chest CT may be difficult in patients with very low body mass index due to a scanty amount of the paracardiac fat. Third, the number of MPE patients was substantially smaller (77 patients) compared with pleural TB patients (351 patients), and there was no attempt to equalize these numbers. Fourth, in patients with MPE, there were various malignant diseases, but no subanalysis was performed based on the type of malignant disease. Fifth, the level of pleural ADA and serum CRP were not measured in a substantial proportion of the patients with MPE in this study. Thus, a comparison of the level of pleural ADA and serum CRP between patients with pleural TB and those with MPE was not performed in this study. Finally, this is a retrospective study based on experience from a single center.

In conclusion, we found that the phlegmonous appearance in the ipsilateral paracardiac fat without paracardiac lymph node enlargement on chest CT favors the diagnosis of pleural TB over MPE.

## Figures and Tables

**Figure 1 diagnostics-10-01041-f001:**
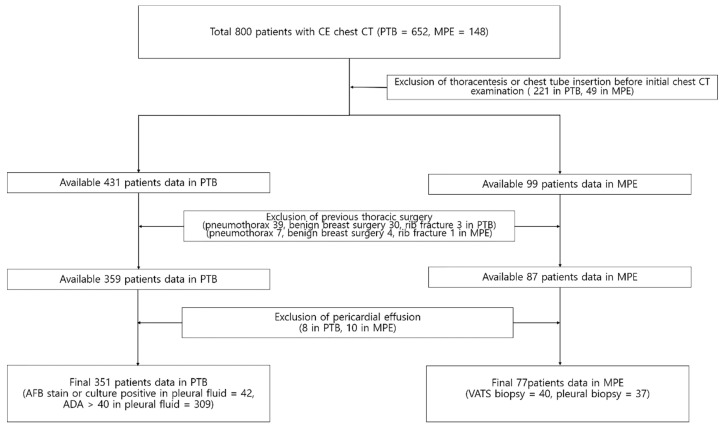
Flow chart showing the selection of the study cohort.

**Figure 2 diagnostics-10-01041-f002:**
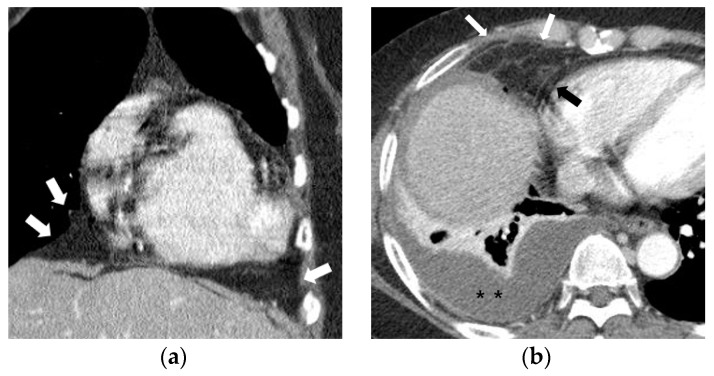
Representative cases demonstrating normal appearance, fat stranding, fluid collection, and phlegmonous appearance in the ipsilateral paracardiac fat on chest computed tomography. (**a**) Normal appearance of paracardiac fat on a coronal CT image at the level of left ventricle. Note a lack of fat stranding or a fluid collection in the paracardiac fat of a patient with pleural tuberculosis (white arrows). (**b**) A case of right pleural tuberculosis (asterisks) in a 67-year-old female with fat stranding (black and white arrows) in the right paracardiac fat on an axial CT image at the level of the right ventricle. (**c**) A case of right pleural tuberculosis (asterisk) in a 77-year-old female demonstrating a fluid collection (white arrows) in the right paracardiac fat on an axial CT image at the level of the left ventricle. (**d**,**e**) Two cases of left pleural tuberculosis (asterisk) with a phlegmon-like appearance in left paracardiac fat in a 74-year-old female on axial CT image (**d**), and in a 58-year-old male on coronal CT (**e**) at the level of the left ventricle. Note multiple fat strands (white arrows) interspersed with multiple fluid collections (black arrows or black arrowheads) in the paracardiac fat ipsilateral to the left pleural effusion.

**Figure 3 diagnostics-10-01041-f003:**
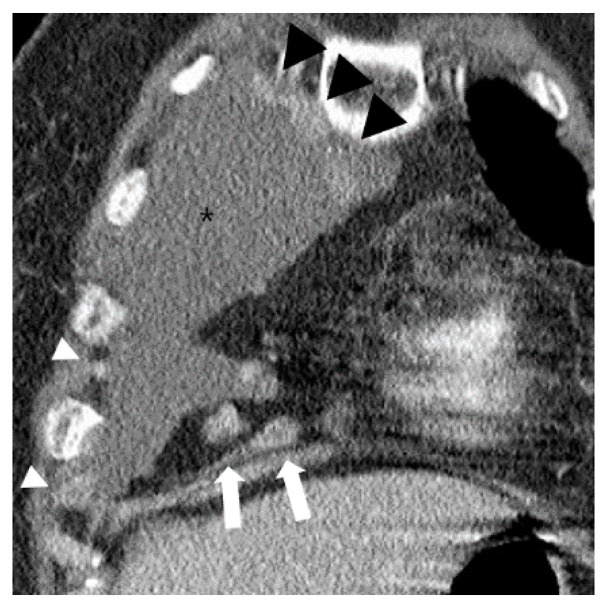
A case of typical malignant pleural effusion (asterisk) in a 73-year-old female on chest CT. Over 1 cm enlarged lymph nodes are noted in ipsilateral paracardiac fat (white arrows). Multiple contrast enhanced pleural nodular thickenings are also noted (white and black arrowheads).

**Table 1 diagnostics-10-01041-t001:** Basic characteristics of patients with pleural tuberculosis and those with malignant effusion.

Demographics	Pleural TB	MPE	*p* Value
Numbers	351	77	
Age (years)	48.1 ± 18 (3–97)	65.1 ± 2 (35–88)	<0.001 *
Gender (male/female)	214 (61)/137 (39)	40 (52)/37 (48)	0.144
Positive AFB stain or culture	42	-	
ADA > 40 (IU/L), pleural fluid	309	-	
VATS biopsy	-	40	
Pleural biopsy	-	37	
Primary malignancy location			
lung cancer		52 (67.5)	
ovary cancer		8 (10.4)	
pleural mesothelioma		5 (6.5)	
malignant pleural solitary fibrous tumor		3 (3.9)	
breast cancer		3 (3.9)	
Gallbladder cancer		1 (1.3)	
Klatskin tumor		1 (1.3)	
esophageal cancer		1 (1.3)	
cervical cancer		1 (1.3)	
thymic carcinoma		1 (1.3)	
peritoneal carcinoma		1 (1.3)	

Pleural TB, Pleural tuberculosis; MPE, Malignant pleural effusion; AFB, Acid fast bacilli; ADA, Adenosine deaminase; VATS, Video-assisted thoracoscopic surgery. Parentheses indicate percentage. * *p* value < 0.05.

**Table 2 diagnostics-10-01041-t002:** Prevalence of phlegmonous appearance, fat stranding, multiple fluid collections, and lymph node enlargement in the ipsilateral paracardiac fat in patients with pleural TB and those with malignant pleural effusion.

Paracardiac Fat	Pleural TB	MPE	OR [95% CI]	*p* Value
Phlegmonous appearance	167/351 (47.6)	8/77 (10.4)	7.8 (3.7–16.8)	<0.001 *
Fat stranding	81/351 (23.1)	18/77 (23.4)	0.9 (0.5–1.8)	0.955
Multiple fluid collections	22/351 (6.3)	6/77 (7.8)	0.8 (0.3–2.0)	0.624
Lymph node enlargement (>1 cm)	5/351 (1.4)	15/77 (19.5)	0.06 (0.02–0.2)	<0.001 *
Phlegmonous appearance without lymph node enlargement	165/351 (47.0)	1/77 (1.3)	67.4 (9.3–490.2)	<0.001 *

Phlegmonous appearance; simultaneous presence of paracardiac fat standing and fluid collection. Pleural TB, pleural tuberculosis; MPE, malignant pleural effusion; OR, Odds Ratio; CI, Confidence interval; Parentheses indicates percentage. * *p* value < 0.05.

**Table 3 diagnostics-10-01041-t003:** Prevalence of CT findings in the ipsilateral paracardiac fat in patients with pleural TB and patients with MPE, lacking typical CT features of MPE (i.e., pleural nodular thickening > 1 cm).

Paracardiac Fat	Pleural TB	MPE	OR [95% CI]	*p* Value
Phlegmonous appearance	166/347 (47.8)	4/45 (8.9)	9.4 (3.3–26.8)	<0.001 *
Fat stranding	79/347 (22.8)	8/45 (17.8)	1.4 (0.6–3.0)	0.449
Multiple fluid collections	21/347 (6.1)	4/45 (8.9)	0.7 (0.2–2.0)	0.464
Lymph node enlargement (>1 cm)	5/347 (1.4)	4/45 (8.9)	0.2 (0.04–0.6)	0.002 *
Phlegmonous appearance without lymph node enlargement	164/347 (47.3)	1/45 (2.2)	39.4 (5.4–289.4)	<0.001 *

Phlegmonous appearance; simultaneous presence of paracardiac fat standing and multiple fluid collections. Pleural TB, pleural tuberculosis; MPE, malignant pleural effusion; OR, Odds Ratio; CI, Confidence interval; Parentheses indicates percentage. * *p* value < 0.05.

**Table 4 diagnostics-10-01041-t004:** Association between level of pleural adenosine deaminase and paracardiac CT findings in patients with pleural TB.

ADA in Pleural TB (IU/L)	Median	Mean ± SD	*p* Value
Phlegmonous appearance			<0.001 *
positive (n = 161)	104	107.1 ± 32.6	
negative (n = 161)	90	94.2 ± 45.1	
Fat stranding			0.43
positive (n = 73)	97.2	97.3 ± 34.6	
negative (n = 88)	83.3	91.6 ± 52.3	
Multiple fluid collections			0.806
positive (n = 21)	96	100.5 ± 71.0	
negative (n = 140)	89.4	93.2 ± 40.1	
Phlegmonous appearance vs. Fat stranding			0.082
Phlegmonous appearance (n = 161)	104	107.1 ± 32.6	
Fat stranding (n = 71)	97.2	97.3 ± 34.6	
Phlegmonous appearance vs. Multiple fluid collections			0.123
Phlegmonous appearance (n = 161)	104	107.1 ± 32.6	
Multiple fluid collections (n = 21)	96	100.5 ± 71.0	

Pleural TB, Pleural tuberculosis; ADA, Adenosine deaminase; Parentheses indicates percentage. * *p* value < 0.05.

**Table 5 diagnostics-10-01041-t005:** Association between serum C-reactive protein and paracardiac CT findings in patients with pleural TB.

CRP in Pleural TB (mg/dl)	Median	Mean ± SD	*p* Value
Phlegmonous appearance			<0.001 *
positive (n = 162)	6.5	8.2 ± 6.8	
negative (n = 172)	4.2	6.8 ± 7.4	
Fat stranding			0.241
positive (n = 76)	4.3	7.1 ± 7.1	
negative (n = 96)	3.8	6.7 ± 7.6	
Multiple fluid collections			0.808
positive (n = 22)	3.9	6.5 ± 7.0	
negative (n = 150)	4.2	6.9 ± 7.4	
Phlegmonous appearance vs. Fat stranding			0.022 *
Phlegmonous appearance (n = 162)	6.5	8.2 ± 6.8	
Fat stranding (n = 76)	4.3	7.1 ± 7.1	
Phlegmonous appearance vs. Multiple fluid collections			0.075
Phlegmonous appearance (n = 162)	6.5	8.2 ± 6.8	
Multiple fluid collections (n = 22)	3.9	6.5 ± 7.0	

Pleural TB, Pleural tuberculosis; ADA, Adenosine deaminase; Parentheses indicates percentage. * *p* value < 0.05.

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
