# Peer review of "Phlegmonous Appearance in the Ipsilateral Paracardiac Fat without Paracardiac Lymph Node Enlargement on Chest CT Favors the Diagnosis of Pleural Tuberculosis over Malignant Pleural Effusion"

_diagnostics, 2020, doi:10.3390/diagnostics10121041_

Round 1
Reviewer 1 Report
The work is original. Despite of it some aspects need to be clarified.
- The authors demonstrated that enlarged lymph nodes in paracardial fat pad without phlegmonous appearance are associated with MPE; this finding is interesting. However in the discussion the authors explain this difference as follows: “This difference may be explained by the fact that mediastinal lymph node enlargement is rare in immunocompetent adult patients with active pulmonary tuberculosis compared to immunocompromised host”; please modify since it is not true that in immunocompetent host there is no lymph node enlargement; moreover in materials and methods the authors do not specify the immune status of patients enrolled with TB. Therefore try to explain the absence of lymphadenopathy in paracardial fat pad in TB patients in a different way.
In the introduction “In many patients lacking prominent nodular pleural enhancement on chest CT, the differentiation of pleural tuberculosis (TB) from MPE can be a challenging task for interpreting physicians [1, 3]”, please add the following references:
- Mazzei MA, Sartorelli P, Bagnacci G et al. Occupational Lung Diseases: Underreported Diagnosis in Radiological Practice. Semin Ultrasound CT MR. 2019 Feb;40(1):36-50;
- Bennett D, Fossi A, Refini RM et al. Posttransplant solid organ malignancies in lung transplant recipients: a single-center experience and review of the literature. Tumori. 2016 Dec 1;102(6):574-581;
- Messina C, Bignone R, Bruno A et al. Diffusion-Weighted Imaging in Oncology: An Update. Cancers (Basel). 2020 Jun 8;12(6):1493.
Author Response
Thank you for your invaluable comments on our article.
The authors demonstrated that enlarged lymph nodes in paracardial fat pad without phlegmonous appearance are associated with MPE; this finding is interesting. However in the discussion the authors explain this difference as follows: “This difference may be explained by the fact that mediastinal lymph node enlargement is rare in immunocompetent adult patients with active pulmonary tuberculosis compared to immunocompromised host”; please modify since it is not true that in immunocompetent host there is no lymph node enlargement; moreover in materials and methods the authors do not specify the immune status of patients enrolled with TB. Therefore try to explain the absence of lymphadenopathy in paracardial fat pad in TB patients in a different way.
-As for this comment, the authors removed the above sentence in the discussion section. Instead, the authors inserted the following sentence in the discussion section:
“However, it is unclear why the prevalence of enlarged lymph nodes over 1 cm in the ipsilateral paracardiac fat is higher in patients with MPE compared to those with pleural TB in this study. Further studies are required to address this issue.”
In the introduction “In many patients lacking prominent nodular pleural enhancement on chest CT, the differentiation of pleural tuberculosis (TB) from MPE can be a challenging task for interpreting physicians [1, 3]”, please add the following references:
-As for the comment, the authors included the recommended articles in the references.
The authors hope you would find the revised article suitable for publication.
Reviewer 2 Report
I have reviewed the paper by Lee et al., on CT parameter associations with either TB or Malignant pleural effusion. The title phlegmonous is rather obscure, as the term was never introduced nor mentioned in the Abstract. It only appears in the second paragraph of 2.3.
There are terms that are inappropriate in the setting of a retrospective study, such as “the incidence of 23 isolated FS or multiple FC within the ipsilateral paracardiac fat between the two groups”.
Exclusion criteria are well thought.
The levels of ADA and CRP should have been measured in MPE cases, in order for this to be considered as a variable. So the purpose of Table 4 and 5 is not clear.
Table 2 summarizes the main findings. There is no mention of Table 3 in the text, so I am puzzled of why Table 3 is presented, as it seems redundant, and an incidence cannot be calculated in a retrospective study. The title says “patients lacking nodular enlargement >1c,” , however the table shows:
Lymph node enlargement (> 1cm) |
5 / 347 (1.4) |
4 / 45 (8.9) |
0.2 [0.04 - 0.6] |
0.002* |
The conclusion “In conclusion, we found that the phlegmonous appearance in the ipsilateral paracaridac fat 261 without paracardiac lymph node enlargement on chest CT favors the diagnosis of pleural TB over 262 MPE.” Is correct. Maybe use that statement as the title.
Author Response
Thank you for your invaluable comments on our article.
I have reviewed the paper by Lee et al., on CT parameter associations with either TB or Malignant pleural effusion. The title phlegmonous is rather obscure, as the term was never introduced nor mentioned in the Abstract. It only appears in the second paragraph of 2.3.
-As for the comment, the authors inserted the term :”phlegmonous appearance” in the abstract as follows:“Two radiologists independently evaluated the chest CT findings regarding the paracardiac fat pad ipsilateral to the effusion including FS, FC, phlegmonous appearance (a combination of the FS and multiple FC), and the presence of lymph node enlargement (> 1cm in short axis diameter).”
There are terms that are inappropriate in the setting of a retrospective study, such as “the incidence of 23 isolated FS or multiple FC within the ipsilateral paracardiac fat between the two groups”.
-As for this comment, the authors used the term “prevalence” instead of incidence in the paper.
Exclusion criteria are well thought.
The levels of ADA and CRP should have been measured in MPE cases, in order for this to be considered as a variable. So the purpose of Table 4 and 5 is not clear.
-As for the comment, the authors inserted the following sentences in the limitation section: “Fifth, the level of pleural ADA and serum CRP were not measured in substantial proportion of the patients with MPE in this study. Thus, comparison of the level of pleural ADA and serum CRP between patients with pleural TB and those with MPE was not performed in this study.”
Table 2 summarizes the main findings. There is no mention of Table 3 in the text, so I am puzzled of why Table 3 is presented, as it seems redundant, and an incidence cannot be calculated in a retrospective study. The title says “patients lacking nodular enlargement >1c,” , however the table shows:
Lymph node enlargement (> 1cm) |
5 / 347 (1.4) |
4 / 45 (8.9) |
0.2 [0.04 - 0.6] |
0.002* |
-As for the comment, the main purpose of Table 3 was to provide the differences in the paracardiac CT findings between the patients with pleural TB andthose lacking typical CT findings of MPE (i.e., nodular pleural enhancement >1 cm).
To remove the potential confusion between the term "lymph node enlargement > 1 cm" and nodular pleural enhancement (>1 cm), the authors revised the heading of Table 3 as follows:
Table 3. Prevalence of CT findings in the ipsilateral paracardiac fat in patients with pleural TB and patients with MPE, lacking typical CT features of MPE (i.e., pleural nodular thickening> 1cm).
In addition, table 3 was mentioned in the materials and methods section in page 7 in the line number of 172~179.
The conclusion “In conclusion, we found that the phlegmonous appearance in the ipsilateral paracaridac fat 261 without paracardiac lymph node enlargement on chest CT favors the diagnosis of pleural TB over 262 MPE.” Is correct. Maybe use that statement as the title.
-As for this comment, the authors used the above sentence as the title of the paper.
The authors hope you would find the revised article suitable for publication.